# Promoting Veteran-Centric Transportation Options through Exposure to Autonomous Shuttles

Sherrilene Classen [1], Isabelle C. Wandenkolk [1,*], Justin Mason [2], Nichole Stetten [1], Seung Woo Hwangbo [1] and Kelsea LeBeau [3]

1 Department of Occupational Therapy, University of Florida, Gainesville, FL 32611, USA; sclassen@phhp.ufl.edu (S.C.); n.e.stetten@phhp.ufl.edu (N.S.); shwangbo@phhp.ufl.edu (S.W.H.)
2 Driving Safety Research Institute, University of Iowa, Iowa City, IA 52242, USA; justin-mason@uiowa.edu
3 Veterans Rural Health Resource Center-Gainesville (VRHRC-GNV), North Florida/South Georgia Veterans Health System, U.S. Department of Veterans Affairs, Gainesville, FL 32608, USA; kelsea.lebeau@va.gov
* Correspondence: iwandenkolk@phhp.ufl.edu

**Abstract:** Veterans face difficulties accessing vital health and community services, especially in rural areas. Autonomous vehicles (AVs) can revolutionize transportation by enhancing access, safety, and efficiency. Yet, there is limited knowledge about how Veterans perceive AVs. This study fills this gap by assessing Veterans' AV perceptions before and after exposure to an autonomous shuttle (AS). Using a multi-method approach, 23 participants completed pre- and post-AS Autonomous Vehicle User Perception Survey (AVUPS), with 10 participants also taking part in post-AS focus groups. Following exposure to the AS, differences were observed for three out of the four AVUPS domains: an increase in *Intention to Use* ($p < 0.01$), a decrease in *Perceived Barriers* ($p < 0.05$), and an increase in *Total Acceptance* ($p = 0.01$); *Well-being* remained unchanged ($p = 0.81$). Feedback from focus groups uncovered six qualitative themes: *Perceived Benefits* (n = 70), *Safety* (n = 66), *Shuttle Experience* (n = 47), *AV Adoption* (n = 44), *Experience with AVs* (n = 17), and *Perception Change* (n = 10). This study underscores AVs' potential to alleviate transportation challenges faced by Veterans, contributing to more inclusive transportation solutions. The research offers insights for future policies and interventions aimed at integrating AV technology into the transportation system, particularly for mobility-vulnerable Veterans in rural and urban settings.

**Keywords:** autonomous shuttle; autonomous vehicle acceptance; intention to use; veterans



## 1. Introduction

Returning combat Veterans may experience significant challenges in accessing essential health and community services, especially those living in rural areas [1]. The scarcity of accessible, convenient, acceptable, economical, and flexible transportation options is particularly pronounced in rural locations. Autonomous vehicles (AVs) represent an emerging technology that has the potential to revolutionize the transportation landscape. The integration of AVs into the transportation system offers numerous potential benefits. Improved road safety is a significant advantage, as AVs can mitigate human error, a leading cause of accidents [2]. Furthermore, AVs have the potential to enhance transportation efficiency, reduce traffic congestion, and optimize fuel consumption through advanced traffic management systems and smoother driving patterns [3]. Nonetheless, the successful integration of AVs requires addressing various challenges and concerns, including establishing public trust and acceptance of AV technology [4]. In particular, little is known regarding Veterans' experiences with, acceptance of, and adoption of AVs, particularly autonomous shuttles (AS). To address this gap, this study exposed rural and urban Veterans to AS technology and subsequently collected their pre- and post-AS experiences. By understanding the factors influencing acceptance of AV technology, this research aims to contribute to the

development of equitable transportation solutions that meet the unique needs of rural Veterans and shape future policies to facilitate the successful implementation of AV technology in the transportation system.

### 1.1. Veterans

Rural Veterans: Improving access for highly rural Veterans, who reside in counties with a population density of fewer than seven individuals per square mile, remains a paramount priority for the Department of Veterans Affairs (VA) [1]. Almost a quarter of all Veterans in the United States (4.7 million) return from active military careers to reside in rural communities, with 58% (2.7 million) of rural Veterans enrolled in the VA health care system [5]. Highly rural Veterans cite distance and transportation as critical obstacles to obtaining care [6]. Indeed, Veterans do not have ubiquitous transportation options meeting the 5A's standards of transportation: accessibility, acceptability, availability, affordability, and adaptability [7].

Urban Veterans: Veterans residing in urban areas face unique challenges when it comes to transportation due to a variety of factors. A subset of urban Veterans opts not to drive due to the effects of Post-Traumatic Stress Disorder (PTSD), a mental health condition that develops in individuals who have experienced or witnessed a traumatic event or series of events. Symptoms associated with PTSD can include flashbacks, nightmares, hyperarousal, and avoidance behaviors. In urban settings, the stresses of heavy traffic and crowded environments can trigger or exacerbate these symptoms [8]. Additionally, a significant portion of urban Veterans cannot operate vehicles due to multiple comorbidities, which make navigating city streets a complex and potentially hazardous endeavor [9,10]. Moreover, combat Veterans frequently encounter challenges in the post-deployment phase as they strive to readapt to civilian driving [11]. This is attributed to the battle-mind tactics acquired during combat, which can intensify their situational awareness to a degree where engaging in urban driving could potentially evoke defensive reactions [8–10]. These responses, in turn, may jeopardize not only their own safety but also that of other individuals on the road. Unlike their rural counterparts, who often face distance-related barriers, these urban Veterans grapple with a distinct set of obstacles stemming from their proximity to high-density living. Ensuring that transportation services align with principles of accessibility, acceptability, availability, affordability, and adaptability, as established by transportation standards, becomes increasingly crucial for addressing the distinct transportation needs of both rural and urban Veterans [7].

### 1.2. Autonomous Vehicles

An autonomous vehicle (AV) refers to a vehicle that can operate and navigate without direct human intervention. These vehicles utilize a combination of advanced technologies, including sensors, cameras, radar systems, and artificial intelligence algorithms, to perceive the surrounding environment, interpret data, and make informed driving decisions [2]. The National Highway Traffic Safety Administration (NHTSA) has adopted the Society of Automotive Engineers (SAE) framework to classify the levels of automation in vehicles, ranging from Level 0 to Level 5. As the level of automation increases, a driver has less responsibility or control over the vehicle. At Level 0, there is no automation, and the vehicle is entirely controlled by a human driver. At Level 1, the vehicle incorporates features like adaptive cruise control or lane-keeping assistance. Level 2 represents partial automation, where the vehicle can control both steering and acceleration (i.e., latitudinal and longitudinal control), but a human driver is still required to monitor the driving environment. Level 3 signifies conditional automation, where the vehicle can manage aspects of driving within its operational design domain but will require human intervention. Level 4 corresponds to high automation, where the vehicle can operate independently in specific conditions or areas (i.e., geofenced), but human intervention is still an option. Finally, Level 5 represents full automation, where the vehicle can operate in any condition without human input [12].

Autonomous vehicles offer numerous potential benefits. Improved road safety is a significant advantage, as they can mitigate human error, which is a leading cause of crashes [2]. AVs also have the potential to increase transportation efficiency, reduce traffic congestion, and optimize fuel consumption through smoother driving patterns and advanced traffic management systems [13]. Additionally, AVs can enhance mobility options for individuals who are unable to drive, such as the elderly or those with disabilities, by providing them with newfound independence and accessibility [2]. However, there are certain drawbacks and challenges associated with AVs. One concern is the issue of liability and the determination of responsibility in the event of accidents involving AVs [2]. Ethical considerations, such as the decision-making algorithms in critical situations, pose another challenge. Ensuring the security and protection of AV systems from cyber threats is also of utmost importance [14]. Moreover, public acceptance and trust in AV technology, along with regulatory frameworks and standards, need to be addressed for the successful integration of AVs into the transportation system [2,4,15].

### 1.3. Autonomous Ride-Sharing Services

As vehicle automation continues to advance, autonomous ride-sharing services (ARSS) have gained significant attention as a potential solution for urban mobility challenges. The ARSS enables travelers with similar or overlapping routes to engage in carpooling, optimizing vehicle utilization, and reducing congestion on the roads [16]. By combining AVs with a ride-sharing model, these services have the potential to enhance accessibility, particularly for underserved populations, by providing convenient and efficient transportation options [17]. Within the field of transportation, the challenge of accessing public transportation is commonly known as the "first- and last-mile problem" [18]. This problem refers to the lack of transit services during the initial and/or final segments of a journey, hindering access to public transit options and leading more individuals to choose private modes of transportation over public ones. However, ARSS hold the potential to function as first- or last-mile service providers, connecting passengers to public transportation hubs or their ultimate destinations [18]. For instance, AVs could pick up passengers from remote locations, potentially including rural areas, and transport them to nearby public transit stations, allowing them to continue their journeys to their final destinations.

By bridging the gap between remote locations and public transit networks, ARSS have the potential to enhance mobility options, reduce reliance on private vehicles, and contribute to a more sustainable and efficient transportation system. However, concerns related to passenger safety, trust in autonomous technology, liability, and regulatory frameworks remain important considerations for the successful implementation of ARSS. Evaluating user experiences, attitudes, and acceptance of these services is essential to understand their potential impacts and identify areas for improvement in terms of service quality, user satisfaction, and overall feasibility.

### 1.4. AV Acceptance Studies

Civilian perceptions and adoption intentions of AV technology vary greatly among sociodemographic groups since commuters value different characteristics of AVs [19]. A pilot test with civilian participants assessed acceptance, perceived safety, trust, intention to use, and emotions following exposure to an AS [20]. The results indicated that participants had statistically significant increases in acceptance, safety, and trust. Although this study demonstrates the promise of AV technology among civilians, Veterans are a unique population that differs significantly from civilians in terms of military-related exposure, training, and health conditions [21]. A separate study investigated sighted and visually impaired Veterans' experiences with AVs using a facial analysis coding system and a user experience questionnaire after exposure to an AS [22]. While the participants responded positively to their experience with the shuttle, this study only assessed their post-exposure perceptions. Post-exposure assessments showed a positive user experience with AS and as such provide valuable insights into the immediate impressions of AV technology. However,

a comprehensive assessment that considers both pre- and post-exposure perspectives is necessary to determine whether exposure to an AS influences users' perceptions of AVs. Notably, no research has specifically examined Veterans' perceptions of AV technology pre- and post-exposure to an AS.

*1.5. Autonomous Vehicle User Perception Survey*

The Autonomous Vehicle User Perception Survey (AVUPS) was developed to quantify travelers' perceptions of and attitudes toward AV technology, particularly in the context of fully AVs (SAE Levels 4 and 5) [23,24]. The survey is rooted in a conceptual model derived from seven influential technology acceptance models: (1) Technology Acceptance Model (TAM) [25], (2) Safety Critical Technology Acceptance Model [26], (3) Car Technology Acceptance Model [27], (4) Unified Theory of Acceptance and Use of Technology [28], (5) TAM-extended framework [29], (6) Self-Driving Car Acceptance Scale [30], and (7) 4P Acceptance Model [31]. The survey items reflect eight sub-dimensions from the conceptual model, which are as follows: (a) intention to use, (b) perceived ease of use, (c) perceived usefulness, (d) safety, (e) trust and reliability, (f) experience, (g) control and driving-efficacy, and (h) external variables such as media, governing authority, social influence, and cost. Via exploratory factor and Mokken scale analysis, the AVUPS yielded three Mokken subscales (i.e., Intention to Use, Perceived Barriers, and Well-Being) and a total score (i.e., Total Acceptance) [24].

*1.6. Rationale and Significance*

In light of the evolving transportation landscape, this study assumes a crucial role in strategically planning future interventions that effectively reintegrate mobility-vulnerable Veterans—those who are either unable to drive or face transportation limitations—into society. The utilization of AS offers numerous advantages, including a reduction in road fatalities and improved accessibility to transportation for individuals who are unable to drive, transitioning away from driving, or facing mobility limitations. Previous research has provided initial insights in studies involving 106 younger and middle-aged adults [32], 16 adults with spinal cord injuries [33], and 42 adults with disabilities [34], who were exposed to the EasyMile (EZ10) low-speed AS. Compared to pre-exposure, all participants demonstrated a positive shift in their perceptions of AS after AS exposure. However, to explore AV technology as a future equitable transportation option for the Veteran population, perceptions, thoughts, beliefs, and values impacting acceptance of the Veterans must be understood.

*1.7. Purpose*

The study assessed the perceptions of rural and urban Veterans regarding AS, both before and after their exposure to the EasyMile (EZ10) AS. To gather Veterans' perceptions, a valid and reliable AVUPS was administered [26,27]. Specifically, we assessed Veterans' perceptions of AVs using three domains (i.e., *Intention to Use*, *Perceived Barriers*, and *Well-Being*) and a total score (i.e., *Total Acceptance*) as the primary outcome variables. By investigating the factors underlying AV acceptance within the context of Veterans' perceptions, the study aims to uncover insights into aspects that are particularly important or relevant to this specific user group. Moreover, since Veterans' perceptions of AS are underexplored and this is a novel technology, focus groups were conducted to delve deeper into the Veterans' perceptions, knowledge, and experiences regarding their exposure to AS.

## 2. Materials and Methods

This study was approved by the University of Florida's Institutional Review Board (IRB), the North Florida/South Georgia Veterans Affairs Research Committee, and the Office of Rural Health. Participants completed an IRB-approved Informed Consent Form (ICF) and could receive up to USD 125.00 if they completed all components of the study.

### 2.1. Design

This study uses a multi-method design, which involves the collection and analysis of quantitative and qualitative data, to assess Veterans' perceptions of AVs before and after exposure to an AS.

### 2.2. Autonomous Shuttle

The study used an EZ10 AS manufactured by EasyMile in Toulouse, France (see Figure 1). The electric shuttle operates autonomously at SAE Level 4, following pre-mapped routes without the need for a steering wheel. Each shuttle was designed to accommodate up to 12 people, providing six seats and a standing area. Notably, the EZ10 shuttle is wheelchair accessible and equipped with an automatic wheelchair ramp. Moreover, the design of the shuttle also caters to users who prefer to use assistive mobility devices, such as walkers or canes. The maximum speed of the shuttle did not exceed 15 MPH on public roads. An on-board experienced safety operator could at any point switch the shuttle from autonomous to manual driving mode using a joystick remote control. The shuttle route (see Figure 2) began at the Southwest Downtown Parking Garage, located at 105 SW 3rd St., Gainesville, FL 32601, USA. The shuttle followed a looped route, with two shuttles operating simultaneously, completing a round trip in approximately 25 min in real-world traffic between the hours of 9 a.m. and 5 p.m. on weekdays. The AS did not operate on days with inclement weather (i.e., heavy rain and/or lightning).

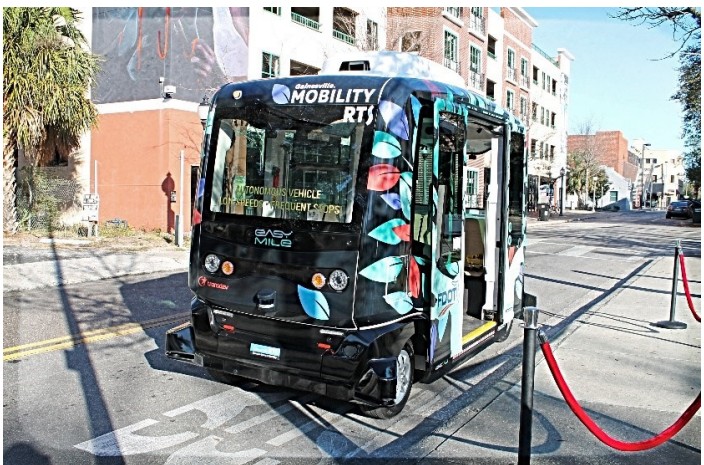

**Figure 1.** EasyMile (EZ10) autonomous shuttle.

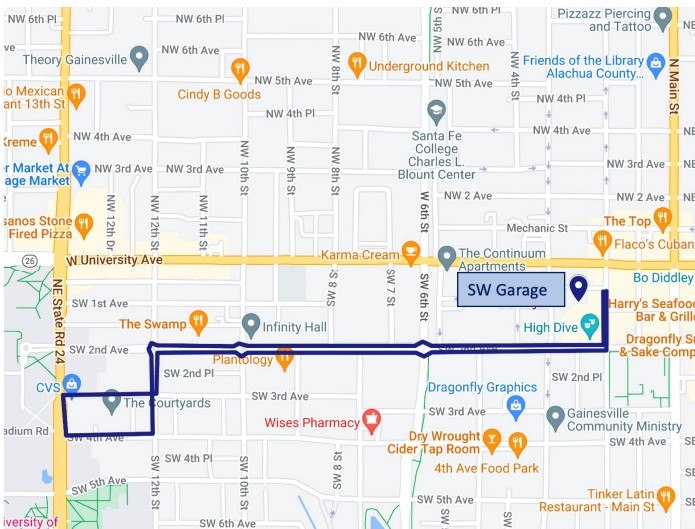

**Figure 2.** Autonomous shuttle route.

*2.3. Participants*

The study team recruited participants via flyer distribution, in-person outreach, and social media platforms. The study team also used the VA Informatics and Computing Infrastructure (VINCI) database to identify and enroll eligible Veterans. The enrollment criteria for the study included Veterans who were 18 years of age or older and proficient in English. Veterans were excluded if they had any medical conditions that impeded them from riding in the shuttle.

*2.4. Measures*

Trained researchers gathered various information from the participants, including their demographics, medical history, and survey responses. The surveys used in this study were the Montreal Cognitive Assessment (MoCA; cognitive screening tool) [35], Technology Readiness Index (TRI) 2.0 [36], Technology Acceptance Model (TAM) [25], Automated Vehicle User Perception Survey (AVUPS) [23,24], and a semi-structured focus group questionnaire.

The research team developed a comprehensive demographic questionnaire with the aim of gathering detailed information about the participants' demographics, military exposure, and mental and physical health history. The demographics covered different aspects such as age, gender, rural or urban living, marital status, health insurance, military background, and health conditions. Recognizing the significance of military exposure as a potentially influential factor, the questionnaire includes a dedicated section aimed at eliciting relevant information about participants' involvement in military service. This section encompasses a range of factors, including the branch of service, duration of service, combat experience, and deployments. Additionally, the latter part of the questionnaire focuses on participants' mental and physical health history, covering an extensive array of inquiries. This section addresses mental health conditions like anxiety, depression, and post-traumatic stress disorder (PTSD), as well as physical health issues such as chronic illnesses, injuries, and disabilities.

The MoCA [35] is a widely used screening tool designed to assess cognitive function. The MoCA assessment consists of various tasks that evaluate different cognitive domains, including attention, memory, language, visuospatial abilities, and executive functions. It takes approximately 10 to 15 min to administer and provides a score out of 30, with a higher score indicating better cognitive performance. The MoCA has shown good sensitivity and specificity in distinguishing between individuals with normal cognition and those with cognitive impairments [35]. It serves as a valuable tool for the detection of mild cognitive impairment (MCI = noticeable cognitive decline that is greater than expected for a person's age but not severe enough to interfere significantly with daily functioning) and signs of dementia.

The TRI 2.0 [36,37] and TAM [25] are tools used to assess individuals' familiarity with and acceptance of technology. The TRI 2.0 evaluates an individual's readiness to use technology across four categories: optimism, innovativeness, discomfort, and insecurity. It consists of 16 items that participants rate on a scale from 1 (strongly disagree) to 5 (strongly agree). For this analysis, only the optimism category was considered, which included four items related to AVs, such as the belief that they contribute to a better quality of life. The TAM includes 26 items that participants rate from 1 (strongly disagree) to 7 (strongly agree). In this analysis, an average score of four items related to ease of use was used [25].

The AVUPS [23,24] is a validated scale used to measure adults' perceptions of AVs. It consists of 28 visual analog scale items, ranging from 0 (disagree) to 100 (agree), and four open-ended questions. The responses to the visual analog scale items are combined to calculate domain scores for *Intention to Use, Perceived Barriers, Well-being,* and *Total Acceptance* of AV technology.

The semi-structured focus group guide was informed by previous research and findings on the AVUPS four open-ended questions that inquired about the perceptions of participants on AV technology across three distinct population categories: young and

middle-aged adults, and people with disabilities [32–34]. The focus groups began with a presentation detailing the five tiers of autonomous driving. The purpose of this presentation was to acquaint the participants with precise terminology and to prepare them for a discussion about any previous encounters they might have had with autonomous technology. Participants were then presented with a series of eight questions, spanning a duration of 30 to 60 min. The scope of these focus groups encompassed a range of topics, including participants' initial perceptions of autonomous technology before riding the shuttle, previous AV experiences, shuttle experience, perception changes in AV technology, likelihood of AV adoption, and the significance of AV in the context of their identities as Veterans.

*2.5. Procedure*

The study was conducted in two phases. Phase I aimed to gather participants' overall perceptions of AV technology by exposing them to the AS. Phase II consisted of focus groups to delve deeper into the participants' experiences riding the shuttle. In Phase 1, Veterans completed various questionnaires before riding the shuttle, including the Demographic Questionnaire, MoCA, TRI 2.0, TAM, and AVUPS. Subsequently, participants took a 25 min ride on the shuttle in downtown Gainesville, FL. The route of the shuttle traversed through a downtown area of the city and included three traffic circles (or roundabouts), seven stops, five right turns, and three left turns while interacting with other road users, pedestrians, cyclists, and background traffic (see Figure 2). Their perceptions of AV technology (dependent variable) were assessed pre- and post-shuttle exposure using the AVUPS. For Phase 2, a qualitative methodologist conducted the focus groups with randomly selected Veterans who had completed Phase 1. The focus group content was recorded and transcribed using VA-Microsoft Teams (Version 1.5, 2022).

*2.6. Data Collection and Management*

Data were collected by the trained research assistant, at the site of the shuttle operation. All records were securely stored either in a locked filing cabinet within a dedicated Veterans' Affairs (VA) research office or on password-protected computers, ensuring compliance with both VA and university information security policies.

*2.7. Data Analysis*

Quantitative data: The data were analyzed using RStudio (PBC, Boston, MA, USA) with R 4.3.0 [38]. Participant demographics were described using frequency (n), proportion (%), mean (M), and standard deviation (SD). To assess the normality of the dependent variables, perceptions of AV technology, visual analysis (histograms and boxplots) and statistical tests (Levene's test, skewness and kurtosis indices, and Shapiro–Wilk test) were conducted. Since the normality assumptions were violated for the dependent variables, a series of Wilcoxon rank-sum tests were performed to assess within-group differences ($p \leq 0.05$) of the four AVUPS domains: *Intention to Use*, *Perceived Barriers*, *Well-being*, and *Total Acceptance*. Descriptive statistics of the AVUPS domains were reported as median and interquartile range (IQR). To control for multiple comparisons (i.e., false discovery rate; $q < 0.05$), the "p.adjust" function in R with the Benjamini–Hochberg method was applied. The associations between Veteran demographics (age; marital status; PTSD; exposure to the primary blast injury, including mortar, Improvised Explosive Device (IED = unconventional/homemade constructed bombs), Rocket-Propelled Grenade (RPG = shoulder-fired anti-tank weapon), grenade, land mine, and/or sniper fire; and injury status), TRI (optimism), TAM (perceived ease of use), and Veterans' *Total Acceptance* of AVs were examined. Furthermore, a post hoc power analysis was conducted to assess the statistical power of the study pertaining to the main outcome variable, *Total Acceptance* of AVs.

Qualitative data: A directed content analysis [39] approach was used to analyze the focus group data from the Veterans. The focus group interview data were analyzed to explore the Veterans' lived experiences and previous qualitative findings by studying a

different sample but with a similar procedure [33]. The analysis took a deductive approach, used within a directed content analysis, and coded based on a priori themes from a previous study [39]. For new codes not within the a priori codebook, an inductive approach was used [39]. During deductive and inductive coding procedures, the constant comparison method was used among three researchers. The coding process was iterative, with researchers refining and revising the codes or categories as new insights emerged. This process continued until data saturation was reached, meaning that no new information surfaced [40].

## 3. Results

### 3.1. Demographics

Table 1 displays the demographic characteristics of the participants at baseline. Out of the 44 individuals screened, we enrolled 23 participants ($M_{age}$ = 55.3, *SD* = 15.8 years), with 19 of them being male. The majority of Veterans fell within the age range of 41 to 64 years (n = 12, 52%). In terms of rurality distribution, most participants resided in urban areas (n = 21, 91%). The participants reported a history of injury (56%) while serving as active-duty members and indicated the presence of PTSD (65%). Most participants reported having health insurance, particularly VA health coverage. Participants were either retired or discharged (96%), and the military branch distribution revealed that the majority of participants were enlisted in the army (43%). Participants reported their marital status as divorced (39%), single (30%), married (17%), or other (13%). Lastly, the average MoCA score was 25 ± 2.8, where a score of 18 to 25 indicates signs of MCI.

**Table 1.** Descriptive statistics for Veterans at baseline.

| Variable | Veterans (N = 23) |
|---|---|
| **Age (years)** | 55.3 ± 15.79 |
| 18–40 | 4 (17.39%) |
| 41–64 | 12 (52.17%) |
| 65+ | 7 (30.44%) |
| **Gender** | |
| Male | 19 (82.61%) |
| Female | 4 (17.39%) |
| **Rural** | |
| Rural | 2 (8.70%) |
| Urban | 21 (91.30%) |
| **Health Insurance** | |
| Yes | 22 (95.65%) |
| No | 1 (4.35%) |
| **VA Health Coverage** | |
| Yes | 20 (86.96%) |
| No | 3 (13.04%) |
| **Marital Status** | |
| Divorced | 9 (39.13%) |
| Single | 7 (30.43%) |
| Married | 4 (17.39%) |
| Others | 3 (13.05%) |
| **Military Branch** | |
| Army | 10 (43.48%) |
| Marines | 6 (26.09%) |
| Navy | 4 (17.39%) |
| Air Force | 3 (13.04%) |
| **Military Status** | |
| Retired/Discharged | 22 (95.65%) |
| Other | 1 (4.35%) |

**Table 1.** *Cont.*

| Variable | Veterans (N = 23) |
|---|---|
| **Live Alone** | |
| Yes/Mostly | 13 (56.53%) |
| No | 10 (43.48%) |
| **Exposure Status** | |
| Yes | 5 (21.74%) |
| No | 18 (78.26%) |
| **Type of Exposure** | |
| Mortar | 4 (17.39%) |
| Improvised Explosive Device | 3 (13.04%) |
| Rocket Propelled Grenade | 3 (13.04%) |
| Grenade | 2 (8.70%) |
| Sniper Fire | 2 (8.70%) |
| Land Mine | 1 (4.35%) |
| **Injury Status** | |
| Yes | 13 (56.52%) |
| No | 10 (43.48%) |
| **Injury Type** | |
| Head | 4 (17.39%) |
| Spine | 5 (21.74%) |
| Arms | 4 (17.39%) |
| Chest | 2 (8.70%) |
| Abdomen | 0 (0.00%) |
| Legs | 6 (26.09%) |
| **Neurologic Disease** | |
| PTSD | 15 (65.22%) |
| TBI | 2 (8.70%) |
| **MoCA Score** | 25.04 ± 2.77 |

Note. Data reported as no. (% of cohort) or mean ± SD. Veterans (N = 23); M = mean; SD = standard deviation.

### 3.2. Correlation Analysis

The research team conducted a correlation analysis between the dependent variable (i.e., *Total Acceptance* post-AS exposure) and the independent variables (i.e., *Total Acceptance* at baseline, age, marital status, exposure status, injury status, PTSD, TRI (*Optimism*) and TAM (*Perceived Ease of Use*)). The results revealed a significant positive correlation ($r = 0.67$, $p < 0.001$) between *Total Acceptance* at baseline and post-AS exposure. Furthermore, a moderate positive correlation between TRI (*Optimism*) and TAM (*Perceived Ease of Use*) was observed ($r = 0.53$, $p < 0.01$). The remaining correlations were not statistically significant and small in magnitude.

### 3.3. The Four AVUPS Scores

Table 2 displays the within-group comparisons of the four AVUPS domains (i.e., *Intention to Use, Perceived Barriers, Well-being,* and *Total Acceptance*) between baseline and post-exposure to the AS. At baseline, the *Intention to Use, Well-being,* and *Total Acceptance* domains displayed scores greater than 65, whereas the *Perceived Barriers* domain scored below 35. Following exposure to the AS, statistically significant differences were observed in all domains, except for *Well-being*. Descriptively, *Well-being* showed an increase post-exposure, although this change did not reach statistical significance. A post hoc power analysis for *Total Acceptance* using the medium effect size observed in this study (Cohen's d = 0.55), a power of 0.80, and an alpha level of 0.05 for matched pairs with two measures (i.e., pre- and post-AS exposure) indicated a sample size of 28 participants. As such, our sample falls short of making definitive statements pertaining to the outcome variables.

**Table 2.** Before and after shuttle exposure: within-group differences in the four AVUPS domains among Veterans.

| AVUPS Domains | Time | | |
|---|---|---|---|
| | Baseline | Post-AS | *p* |
| Intention to Use | 70.08 (27.58) | 83.23 (28.58) | **0.006** |
| Perceived Barriers | 34.50 (30.67) | 23.50 (30.00) | **0.013** |
| Well-being | 72.00 (24.50) | 79.75 (31.00) | 0.808 |
| Total Acceptance | 65.85 (27.45) | 80.65 (28.35) | **0.010** |

Note. Veterans (N = 23); AS = autonomous shuttle; M = median (IQR = interquartile range); Wilcoxon rank-sum test before and after the shuttle (i.e., within-group differences). Significant results in the table are bolded.

*3.4. Focus Group Data*

Table 3 illustrates the qualitative themes, subthemes, and their respective operational definitions and participants' quotes. A directed content analysis revealed six major themes and seven subthemes. The six qualitative themes that have emerged from the data collected from the focus groups include *Perceived Benefits, Safety, Experience with AV, Shuttle Experience, AV Adoption,* and *Perception Change*. Table 4 presents frequency counts for each major theme, offering a clear visualization of the prominence of certain themes within the focus group discussions. These frequency counts, determined by the number of times a theme was mentioned by participants, show the emergence of *Perceived Benefits* (n = 70), *Safety* (n = 66), and *Shuttle Experience* (n = 47) as the top three themes. Following closely in frequencies are *AV Adoption* (n = 44), *Experience with AV* (n = 17), and *Perception Change* (n = 10).

**Table 3.** Qualitative themes, subthemes, and their respective quotes.

| Themes/ Sub-Themes | Definitions | Quotes |
|---|---|---|
| **Perceived Benefits** | Individual's perception of the usefulness of AVs. It includes factors such as the perceived value, benefits, and advantages of using AVs over traditional vehicles. | "We were in some heavy traffic. He started playing Tic Tac toe." "You know you could be at a stadium and call your car to come get you." "Taking the vehicle to different locations and then not having to worry about finding parking spots." "I could take a nap on the way down there." "Well, one thing it saves energy. it's economical if I'm if I'm going to a meeting or something that give me time to look over my papers, you know?" |
| *Perceived ease of use* | Individual's perception of the effort required to use AVs. It includes factors such as perceived complexity, ease of learning, and ease of interacting with the technology (user-friendly). | "It was easy." "Very user friendly." "And even had a little map, screen for you to follow where you were." |
| *Availability* | Access to AVs includes the availability of AVs in the local area or access to AV services/providers. Adequacy of infrastructure to support AV usage, including availability of charging stations and support systems for maintenance and repairs. | "They need it in a lot of the places like Gainesville, Ocala, especially around the Veterans hospitals." "That would make it a lot easier for people like I was saying for like in big huge parking lots where you have to park way out and you walk about." "I only wish that it was more widespread." |
| *Accessibility* | The consideration of diverse user needs, including individuals with disabilities, elderly users, or users with varying technological literacy, and the provision of accessible features or accommodations in AVs. | "It would save me a lot of the walking too because it's hard for me to get around." "I mean, you know, I'm getting up in age, so it would probably help me a lot, you know. As you age, your motor skills decrease, so I'll still have a way of it's a way of getting round." "Yes, you know, people with disabilities you still can be mobile with your disability." |

Table 3. *Cont.*

| Themes/ Sub-Themes | Definitions | Quotes |
|---|---|---|
| **Safety** | Individual's perception of the safety of AVs. It includes factors such as the perceived risks, hazards, and potential accidents associated with AVs. | "It's just the self-driving and getting the person to their destination safely that I really emphasize." "It needs to be safe." "I felt fairly safe." "Safety is the number one concern." |
| *Trust and reliability* | Participants' perceptions of the trustworthiness and dependability of AVs. It includes aspects such as participants' confidence in the technology's ability to navigate safely, the reliability of the vehicle's performance, and their trust in the system's ability to operate safely and effectively in various driving scenarios. | "I had No Fear of it whatsoever." "Well it has all the safety precautions built in. You know, if you got too close to something, it would stop or you know, it gave signals, you know, and and it had all the audio that let you know what's going on, you know." "It's probably more reliable than human." |
| **Experience with AV** | Individual's actual experience with AVs. It includes factors such as the individual's past interactions with AVs and the feedback received from other users. | "I don't think I've had any experience with a lot of automatic, you know driverless vehicles or anything, actually." "My previous experiences are that I have a brother in Tampa who has a Tesla." "I don't remember which airport, but one of the airports I was at about a year ago had an autonomous shuttle." |
| **Shuttle Experience** | Participants' experiences specifically related to using the study's autonomous shuttle. It includes aspects such as the ease of boarding and disembarking, the overall efficiency of the shuttle system, and any notable positive or negative experiences encountered during their shuttle rides. | "Yeah, I like it. It is very neat and efficient. I think it's going to be a good vehicle." "I was worried about, like when another car would come up close and what it would do, you know, and then it handled it pretty good, you know, goes around traffic, it stops when it sees something." "It was the one street that it was all like there was work on the middle of the street. So it had to, you know, stop and wait for the workers to get out of the way and then go around. You know, it was pretty interesting how it did that." "Less operator assistance." "The operator was professional and and answered all my questions and you know it was a very pleasant experience." |
| *Comfort* | Participants' perceptions of comfort while using the autonomous shuttle. It includes their feelings of physical comfort (e.g., seat comfort, vehicle ergonomics) as well as psychological comfort (e.g., feeling safe, relaxed, or confident) during the shuttle ride. | "It was comfortable." "The only other thing that I didn't like were the seats. They were just like regular hard bus seats." "You could fit probably 6 people very comfortably in it, plus a few standing locations too." |
| *Speed* | Participants' perceptions of the speed of the autonomous shuttle. It includes their opinions on the vehicle's acceleration, deceleration, and overall speed during the ride. Participants' experiences with the vehicle's speed in relation to their expectations or preferences. | "I was hoping it was gonna pick up speed." "Going so slow it might actually cause an accident because a lot of times impaired or just drivers that don't pay attention will be expecting to continue at a standard flow, and the autonomous shuttle seems to be a little slower than that." "My biggest concern was speed and time." |
| **AV Adoption** | Participants' inclination or readiness to adopt and utilize AVs in the future. It encompasses their expressed intentions, plans, or willingness to use autonomous vehicles for their transportation needs. Participants' motivations, barriers, and factors influencing their intention to use autonomous vehicles. | "I am here because I am fascinated with self-driving vehicles." "It would be something I would use regularly because I do go to the VA several times a week. And so if I didn't have to worry about where to park and things like that, then I definitely would use it more." "I would absolutely use it." |

**Table 3.** *Cont.*

| Themes/ Sub-Themes | Definitions | Quotes |
|---|---|---|
| *External variables* | External factors that may influence the adoption and use of AVs. These factors may include media coverage, governing authority regulations, social influence, and cost. | "If I could afford one, I would buy one." "The only thing I've seen on advertising is these cars that parked themselves, you know, pull up to a real short space and just the wheels turn and then they just slide into it." "I did do some research on autonomous vehicles and there are several states that actually are ready are implementing the tractor, trailer, truck driving autonomous." |
| **Perception Change** | Perception change refers to the shift in individuals' beliefs, attitudes, or perspectives related to autonomous vehicle technology as a result of their exposure, experience, or knowledge acquisition. It involves the transformation of preconceived notions, biases, or initial impressions about autonomous vehicles into new understandings or perspectives. | "Changed my mind." "Improved my perception a little bit." "I have not changed my opinion." "I'm leaning more towards it for it then against it. I was already more for it. I'm even more for it now." |

**Table 4.** Total frequency counts of each major theme.

| Theme | Frequency Counts |
|---|---|
| Perceived Benefits | 70 |
| Safety | 66 |
| Experience with AV | 17 |
| Shuttle Experience | 47 |
| AV Adoption | 44 |
| Perception Change | 10 |

The Perceived Benefits: The overarching theme of *Perceived Benefits* encompassed three subthemes: perceived ease of use, availability, and accessibility. Through qualitative analysis of participant responses, it is evident that the AS service is regarded as an effortless mode of transportation, contributing to its perceived ease of use. Furthermore, participants expressed the potential for substantial community benefits if shuttle routes were expanded, emphasizing the aspect of availability. This expansion, participants believed, could significantly enhance accessibility, particularly for healthcare providers and an aging and disabled population. Another noteworthy advantage highlighted by participants was the freedom to engage in multitasking during the shuttle ride, and relieving concerns about parking.

Safety: The theme, *Safety*, included one subtheme, trust and reliability, which explored the complex dynamics involved in participants' initial concerns and subsequent development of trust and confidence in the safety of AS. Prior to their experience with the shuttle, participants had concerns about the safety of the AS. However, following their first-hand encounters, many participants conveyed their astonishment at the shuttle's adherence to safety standards and cautious operations. As the participants gained greater familiarity with the AS, they continuously expressed that repeated engagements with its operations and features would strengthen their trust and confidence in its reliability.

Experience with AV: Participants discussed their previous experiences with AVs. Some participants had experience with AVs, such as airport shuttles or Teslas, while others had no prior experience with AVs. In instances when participants lacked prior exposure to AVs, they had acquired knowledge about the technology through different media channels facilitating their understanding of AVs.

Shuttle Experience: The theme, *Shuttle Experience*, included two subthemes: comfort and speed. While some participants described a positive and comfortable experience when

riding in the AS, others expressed discomfort with the (hard) seats. The biggest concerns surrounded the (slow) speed and harsh braking of the shuttle.

AV Adoption: The theme, *AV Adoption*, included one subtheme: external variables. Participants delved into various factors that could sway their preference towards utilizing AS instead of their private cars. Among the multitude of external factors cited by participants, cost emerged as a prominent influencer. The theme of *AV adoption* was often intertwined with the theme of *Perceived Benefits* and *Safety*. Participants consistently conveyed that their inclination to regularly use an AS depends largely on its safety, and availability with routes that conveniently connect with VA hospitals.

Perception Change: Participants discussed how their perceptions might have changed before and after riding in the AS. The majority of participants were already favorably positioned toward AVs, and their exposure to the shuttle helped reinforce or validate their beliefs. In contrast, some participants held initial reservations prior to their shuttle experience; however, these reservations dissipated after the ride, indicating a potential improvement in their perceptions.

## 4. Discussion

This study examined the perceptions of Veterans prior to and after exposure to the EasyMile EZ10 AS. Specifically, using quantitative methodologies, we explored the relationship between the dependent variable, *Total Acceptance* post-AS, and various independent variables. Likewise, using qualitative analysis, we conducted semi-structured interviews to explore deeper insights into the participants' perceptions, emotions, and beliefs concerning AVs. The qualitative data provided more insights into the Veterans' perceptions; however in-depth exploration was limited due to the small sample sizes in the quantitative and qualitative data. Still, the qualitative findings did demonstrate initial apprehensions, shifts in attitudes after exposure, and the underlying factors influencing the Veterans' resistance or acceptance towards AS technology.

### 4.1. Demographics

The sample was predominantly middle-aged to older male Veterans. This demographic aligns with the current Veteran population demographics in the United States [41]. While the majority of participants reside in urban areas (91%), approximately 25% of Veterans live in remote (i.e., residing more than 60 min from the nearest VA) or highly rural areas (i.e., fewer than seven persons per square mile) [1]. Thus, the study has an underrepresentation of rural Veterans. Ironically, their rurality may have reduced the likelihood of being recruited to participate in this study. A significant proportion of participants reported a history of injury while serving as active-duty members (56%) and indicated the presence of PTSD (65%). These figures may suggest that the study's sample included a relatively high percentage of Veterans with injuries and PTSD compared to the general combat Veteran population [42].

Most participants reported having health insurance, particularly VA health coverage, which aligns with the principle that Veterans have access to VA healthcare benefits [43]. The army was the predominant branch among the study participants (43%), aligning with the current Veteran population demographics [41]. Most participants reported their marital status as divorced or single, contrasting with the general Veteran population, where being married is more prevalent [41]. Lastly, the average MoCA score indicated signs of MCI, which is not totally surprising given the presence of older age, and the participants' history of combat exposure and head injuries, which may at least partially account for this phenomenon.

### 4.2. Correlation Analysis

The correlation results indicate a significant positive correlation between *Total Acceptance* at baseline and post-exposure to the AS. This correlation suggests that individuals with a higher initial level of *Acceptance* were more likely to maintain or increase their

*Acceptance* after exposure. Notably, despite the AS technological (e.g., inability to find a signal for GPS coordinates, which inhibited the ability of AS to run autonomously), operational (e.g., schedule changes that impacted participants' opportunities to ride the AS) and weather-related challenges (e.g., rain or gust winds that prohibits the AS from running), participants were overall positive as measured in the increase in *Acceptance* post-exposure. Furthermore, a positive correlation was observed between TAM (perceived ease of use) and TRI (optimism), indicating that participants with higher levels of perceived ease of use also demonstrated elevated optimism levels, and vice versa. This suggests that those who found it effortless and convenient to operate and use AVs were more optimistic about the positive impact of these vehicles on their lives; conversely, those with a more optimistic outlook found the usage of AVs to be easier to use and more convenient. Autonomous vehicle pre-deployment strategies can be tailored to those groups who do not initially think the shuttle will be easy to use or who do not demonstrate optimism. The remaining correlations were not statistically significant and exhibited moderate to small effect sizes, possibly due to the limited sample size and the potential of Type 2 error. Therefore, further investigation with a larger sample, which is adequately powered for each one of the outcome variables, is warranted.

### 4.3. The Four AVUPS Scores

At baseline, the domains of *Intention to Use, Well-being*, and *Total Acceptance* displayed scores above 65 (out of 100), indicating a generally favorable attitude toward AVs. Meanwhile, the scores for the *Perceived Barriers* domain were below 35 (out of 100), suggesting a low perception of obstacles. This indicates that the majority of participants already held a positive view of AVs before riding the shuttle. This positive orientation may be understood in the context of the study's participants, as individuals who are opposed to the implementation of AVs might have been less likely to participate in the study. However, this also suggests that self-selection bias may be inherent to the study.

After being exposed to the AS, participants increased their perceptions of the *Intention to Use* and *Total Acceptance* domains. Moreover, exposure to AVs led to a decrease in participants' perceptions of obstacles, as measured by *Perceived Barriers*, related to using and accessing AVs. This suggests that the practical experience of interacting with AV technology, through the shuttle ride, positively influenced participants' perceptions, making them more receptive to AVs as a viable transportation option. These findings are consistent with prior research exploring AV users' perceptions and attitudes, which also reported an increase in positive attitudes and acceptance of AVs after experiencing the technology firsthand [32,33]. This consistency reinforces the idea that real-world exposure to AVs can play a critical role in shaping members of the public's attitudes and willingness to potentially adopt this technology.

On the other hand, the *Well-being* domain remained stable after the shuttle ride. A previous study utilizing the AVUPS also reported similar results, as *Well-being* did not exhibit any significant changes [33]. Riders may require more exposure to AVs and AS or different use cases if they are to perceive benefits to their well-being (i.e., riding the shuttle on their daily commute or as part of their transportation to and from appointments). This discrepancy raises the possibility of a potential measurement flaw in the *Well-being* domain. To ensure the validity of future studies and to better understand the relationship between AVs and users' well-being, further investigation into the *Well-being* domain of the AVUPS is warranted.

With a sample size of only 23 participants, the study falls short of the recommended 28 participants for detecting a significant difference in *Total Acceptance*, with a medium effect size, 80% power, and a 0.05 alpha level. Although the study did show a statistical significance in *Intention to Use, Perceived Barriers*, and *Total Acceptance*, which is consistent with prior research [33,34], results need to be interpreted with caution. Additionally, the study may have been underpowered to detect smaller yet potentially significant differences. The small sample size further limits the generalizability of the study's findings to a broader

population. Future studies need to have an adequate sample for internal and external validation of study results and for generalizability.

### 4.4. Focus Group Data

Interestingly, the focus group themes reveal a similarity to deductive codes identified in a previous study on AS perceptions. The parallels between the deductive codes and the subthemes found in this study, as demonstrated in Table 3, underline the robustness and consistency of the emerging themes across different research contexts.

Participants' most common theme was *Perceived Benefits*. Participants were quite optimistic about the potential benefits and usefulness of AVs. Participants viewed the AS as a convenient and straightforward mode of transportation. This perception aligns with the concept of user-friendly design, which can play a pivotal role in fostering positive attitudes and encouraging adoption. Participants also underscored the potential for broader community benefits if the shuttle routes were expanded, which resonates with the notion of addressing transportation gaps, particularly in underserved areas. Notably, participants highlighted the positive impact of expanded routes on accessibility, particularly for vulnerable populations such as the elderly and disabled, thereby underscoring the potential for societal inclusivity.

*Safety*, the second most common theme, delved into the intricate interplay between initial concerns and the development of trust and confidence in AS *Safety*. Participants initially harbored reservations about the safety of AS. However, their first-hand experiences led to a shift in perception. Participants were pleasantly surprised by the shuttle's meticulous safety measures and cautious operation, leading to the gradual establishment of trust. This progression aligns with the psychological process of building trust via direct exposure and familiarity. The theme highlights the transformative power of experiential learning in shaping attitudes toward AV technology.

*Shuttle Experience*, the third most common theme, encompassed participants' reflections on their firsthand encounters with the AS. While some participants reported positive and comfortable experiences during their rides, concerns were voiced about certain aspects of the experience. Specifically, discomfort with the seating and dissatisfaction with the speed and braking of the shuttle were apparent. These responses emphasize the significance of ensuring passenger comfort and addressing concerns related to speed control and braking mechanisms to enhance the overall shuttle experience.

As such, the focus group data highlight the potential for AVs to positively impact transportation, particularly for vulnerable populations; the significance of experiential learning in shifting perception; and the need for a user-centric approach to address comfort and safety concerns.

### 4.5. Limitations

The characteristics of the sample may indicate self-selection bias in a convenience sample from North Central Florida. The small sample size may potentially have given rise to Type 2 errors, meaning that an effect could have existed (in the correlations) but was not detected due to small sample sizes. The limited sample size also hindered the recruitment of focus group participants; thus, integrating the quantitative and qualitative data is not feasible. The technology is still in the developmental phases, and we have experienced many issues in the field pertaining to the execution of the study. For instance, the weather tolerance of the shuttles is restricted; they can only function in light rain, but heavy rain hinders the operation of the AS. Moreover, during the summer season, air conditioning consumes a substantial amount of battery power, which can become problematic and may necessitate the shuttle to suspend its operations temporarily for recharging. Lastly, technical issues with shuttle reboots have resulted in difficulties during the turning-on process, leading to the rescheduling of participants and causing delays in successfully completing the study. Although the shuttle provides a first mile–last mile option, it is

running on a fixed route and may not yet provide the accessibility that is needed for Veterans to reintegrate back into their communities.

*4.6. Strengths*

Technology-based interventions, such as deploying the AS as a mode of community mobility, may be considered as a future option for Veterans, especially among those who do not want to drive, cannot drive any longer, or should not be driving [9]. These autonomous shuttles are being deployed throughout the world, and this study is one of the first to identify the perceptions of Veterans pertaining to accepting and potentially adopting these technologies. We have used state-of-the-art technology, collaborators, industry partners, and community involvement to conduct this study; as such, this study reflects the lived experience of the Veterans pertaining to AS. Furthermore, the qualitative findings provide a more detailed narrative that enhances our understanding of the interplay between the individual's narratives in the focus groups and their overall acceptance of AVs as measured in the qualitative data analysis.

## 5. Conclusions

Overall, these results suggest that participants in this study exhibited favorable attitudes toward the AS at baseline, and after exposure to the shuttle pertaining to *Intention to Use*, *Perceived Barriers*, *Well-being*, and *Total Acceptance*. The findings highlight the importance of initial *Acceptance* and individual factors, such as optimism and perceived ease of use, in shaping acceptance. Additionally, participants overall appraised the shuttle experience as positive and comfortable. They expressed strong interest in adopting the AS in their personal vehicles, provided that it offers convenience and increased availability, particularly with additional routes servicing the VA hospital. These results contribute to the existing literature on AVs and provide valuable insights into the design and implementation of AS programs in similar populations. The limitations of this study include its small sample size, which impacted the generalizability and robustness of our findings to a broader population. Further research is warranted to explore additional factors that may influence acceptance and to validate these findings in a larger sample with a more diverse population, and in a more expansive geographic area.

**Author Contributions:** Conceptualization, methodology, funding acquisition and resources, S.C.; validation, S.C., J.M. and N.S.; formal analysis, I.C.W., J.M., N.S., K.L. and S.W.H.; investigation, I.C.W., N.S. and K.L.; data curation, I.C.W.; software, J.M. and I.C.W.; writing—original draft preparation, I.C.W. and S.C.; writing—review and editing, N.S., J.M., K.L. and S.W.H.; visualization, I.C.W.; supervision and project administration S.C., N.S. and J.M. All authors have read and agreed to the published version of the manuscript.

**Funding:** This research was funded by the VA Office of Rural Health (Project: P0213747).

**Institutional Review Board Statement:** The study was conducted in accordance with the Declaration of Helsinki and approved by the Institutional Review Board of the University of Florida (ID: 202101463; approval date: 12/2021).

**Informed Consent Statement:** Informed consent was obtained from all subjects involved in the study.

**Data Availability Statement:** Data sharing not applicable.

**Acknowledgments:** University of Florida's Institute for Driving, Activity, Participation, & Technology; Principal Investigator, Sherrilene Classen; North Florida/South Georgia Veterans Health System; the Malcom Randall VA Medical Center; and the Gainesville Office of Rural Health for infrastructure use and support.

**Conflicts of Interest:** The authors declare no conflict of interest. The funders had no role in the design of the study; in the collection, analyses, or interpretation of data; in the writing of the manuscript; or in the decision to publish the results.

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
