# Peer review of "Promoting Veteran-Centric Transportation Options through Exposure to Autonomous Shuttles"

_safety, 2023_

Round 1

Reviewer 1 Report

Comments and Suggestions for Authors

This study has examined Veterans’ AV perception before and after exposure to AS, using a questionnaire-based survey and a focus group. I agree that Veteran group deserves more research attention to improve their acceptance towards AVs, however, the specific contributions of this study were not clearly stated. My specific comments are listed below.

1. In Introduction, the authors have effectively emphasized the importance of understanding Veterans' perception and acceptance of AVs. However, it would be beneficial if they reviewed previous studies that have explored factors influencing AV acceptance. By doing so, they can clarify how their research contributes to the existing body of knowledge. For instance, did this study investigate previously overlooked factors or unique characteristics of Veterans that might affect AV acceptance differently? The authors should highlight the theoretical contributions of their work in this context.

2. The sample size used in this questionnaire-based study appears to be relatively small. It would be advisable for the authors to address how this sample size might affect the generalizability and robustness of their findings. Discussing the limitations of a small sample size and potential ways to mitigate its impact on the study's conclusions could enhance the credibility of their research.

3. The study identifies some significant correlations, such as the relationship between total acceptance at baseline and post-exposure to AS. These findings fall into common sense and therefore the authors need to discuss how these expected results contribute to the overall understanding of AV acceptance, and whether they reveal any specific insights that can guide future research or real-world AV implementation strategies.

4. The study also found some insignificant correlations, what are kind of unexpected but could be potentially valuable. However, it's important to note that the study's small sample size, as acknowledged by the authors, limits the ability to draw definitive conclusions from these insignificant correlations.

5. The inclusion of a focus group after the survey is somewhat unclear in terms of its purpose and the contribution of its results. The authors should clarify why the focus group was conducted following the survey and elaborate on how the focus group findings inform or complement the survey results. A more explicit connection between the two data collection methods would provide a stronger rationale for including both in the study.

Reviewer 2 Report

Comments and Suggestions for Authors

Thanks for submitting this paper for being considered in Safety. The manuscript (safety-2614481) is empirical study investigating AVs' potential to alleviate transportation challenges faced by Veterans. The topic addressed is worth of investigation, and the theoretical/empirical approaches considered result interesting and may contribute in a substantial manner to the scientific state of Veterans’ mobility. The manuscript is easy to follow, the statistically analysis is reasonable, the conclusion is accurate. Therefore, I believe that the article is ready for publication without any modifications.

Reviewer 3 Report

Comments and Suggestions for Authors

This study is developed to explore the potential for veterans to use autonomous shuttles as another transportation option. Surveys were designed and response results showed that in general, veterans exhibited a favorable attitude toward the AS at baseline and after exposure to the shuttle pertaining to Intention to use, perceived barriers, well-being, and total acceptance. Overall, the study results are interesting and I only have a few minor comments:

1) Definitions of some terms e.g., PTSD, IED, RPG, MCI, MoCA, etc. please define them at their first use;

2) 10,43% ->10.43%

Reviewer 4 Report

Comments and Suggestions for Authors

The topic of the article is relevant. However, the structure of the article itself is not typical for a scientific article.

The problem under consideration is discussed in the Introduction. Some of the material is of a very general, descriptive nature. The second chapter describes the material and methods. Not very structured, but more descriptive material is presented. Two Figures presented in this chapter also of a more general nature: one shows a vehicle, the other a route.

The research results are presented in the third chapter. Some of the results are simply general statistics (Table 1). The information presented in Table 3 is not very specific to the research, but is more descriptive in nature. In my opinion, the scientific originality and novelty of the research are not sufficient. And the research methods and tools used are very simple. Therefore, the discussion and conclusions are also more general in nature. In my opinion, the scientific level of the article is not sufficient.

Round 2

Reviewer 1 Report

Comments and Suggestions for Authors

Thanks for addressing my comments. I have no further questions.

Reviewer 3 Report

Comments and Suggestions for Authors

No further comments.

Reviewer 4 Report

Comments and Suggestions for Authors

Thanks to the Authors for the corrections! But no substantial corrections have been made according to my comments. In my opinion, the scientific level of the article is not sufficient.